# Single-Frequency Ring Fiber Laser with Random Distributed Feedback Provided by Artificial Rayleigh Scattering

Mikhail I. Skvortsov [1,*], Kseniya V. Proskurina [1], Evgeniy V. Golikov [1], Alexander V. Dostovalov [1], Alexey A. Wolf [1], Zhibzema E. Munkueva [1,2], Sofia R. Abdullina [1], Vadim S. Terentyev [1], Olga N. Egorova [3], Sergey L. Semjonov [4] and Sergey A. Babin [1]

[1] Institute of Automation and Electrometry, Siberian Branch of the Russian Academy of Sciences, 1 Ac. Koptyug Ave., 630090 Novosibirsk, Russia; ksyna_98@mail.ru (K.V.P.); golikov.inc@mail.ru (E.V.G.); dostovalov@iae.nsk.su (A.V.D.); wolf@iae.nsk.su (A.A.W.); z.munkueva@g.nsu.ru (Z.E.M.); abdullinasr@iae.nsk.su (S.R.A.); terentyev@iae.nsk.su (V.S.T.); babin@iae.nsk.su (S.A.B.)
[2] Department of Physics, Novosibirsk State University, Pirogova 2, 630090 Novosibirsk, Russia
[3] Prokhorov General Physics Institute of the Russian Academy of Science, 38 Vavilov St., 119991 Moscow, Russia; egorova@nsc.gpi.ru
[4] Dianov Fiber Optic Research Center, Prokhorov General Physics Institute of the Russian Academy of Science, 38 Vavilov St., 119991 Moscow, Russia; sls@fo.gpi.ru
* Correspondence: skvorczov@iae.nsk.su

**Abstract:** Femtosecond (fs) laser inscription technology allows for the production of in-fiber disordered structures with an enhanced level of Rayleigh backscattering with relatively few induced losses. These properties enable the application of these structures as reflectors in fiber lasers. In this study, a narrow-linewidth erbium fiber laser with random distributed feedback provided by a fs-induced random structure in a ring cavity configuration was developed. A single-frequency regime was observed over the entire lasing power range. At a maximum output power of 7.8 mW, the linewidth did not exceed 0.75 kHz.

**Keywords:** single-frequency fiber laser; random distributed feedback; femtosecond refractive index modification

## 1. Introduction

Random distributed feedback (RDFB) fiber lasers based on disordered structures are of great interest nowadays. Due to the ease with which they can be implemented, the application of such high-power narrow-linewidth sources has been shown to be relevant in many areas, ranging from interrogation [1] and remote sensing [2,3] to nonlinear frequency doubling [4,5]. The first proceedings on random fiber lasers were dedicated to random Raman fiber lasers (RFL) with RDFB due to Rayleigh backscattering in natural fluctuations of the refractive index of the optical fiber [6]. The distinctive feature of such lasers is their length, which is of the order of hundreds of meters or hundreds of kilometers. A typical generated linewidth ranges from 1 to 2 nm [6].

Such Rayleigh RDFB provided by long passive fibers (L~1–10 km) is also implemented in fiber lasers applied to active rare-earth-doped fibers. In [7], an erbium fiber laser in a ring cavity configuration with a 5 km passive fiber providing RDFB was presented. Dual-wavelength generation was achieved due to the pair of fiber Bragg gratings (FBGs) with different resonant wavelengths as spectral selection elements. For both wavelengths, single-frequency operation was obtained with a linewidth of ~1 kHz.

Since the amplitude of Rayleigh backscattering for standard optical fibers is extremely low, a great deal of attention is paid to the enhancement of distributed feedback that can be realized through, in particular, the inscription of artificial reflectors, which are continuous or consist of discrete points or separated planes, that allow for a significant reduction in random laser cavity length. In short Raman lasers (L~0.5~10 m), RDFB is realized through

the inscription of a long FBG with a multitude of random phase shifts along the grating [8] or via the inscription of the grating array with random distances between adjacent FBGs [9]. Single-frequency generation regimes can be achieved in certain output power ranges, but such sources demonstrate low stability because of mode competition and thermal effects.

In a recent work [10], an erbium fiber laser in a ring cavity configuration utilizing artificial RDFB was presented. The RDFB structure presented an FBG array with a reflectivity of about 93.5% consisting of eight 0.75 mm FBGs separated by non-modified pieces of fiber with random lengths of 2–8 cm. The low generation threshold and narrow linewidth near the threshold (~0.4 pm) were explained by the Anderson localization effect in the RDFB structure that led to a significant improvement in the laser quality factor.

Artificial RDFB can be realized directly in an active medium. Thus, in [11], a ytterbium laser based on a 4 m FBG consisting of 10 subgratings with random phases and amplitudes of refractive index modulation inscribed in an active fiber was presented. The presented RDFB laser generated a single longitudinal mode with an output power up to 25 mW, and at this level, the linewidth was measured to be <100 kHz. Also, in [12], an erbium-doped fiber laser based on a 5 m active fiber comprising an array of weak FBGs with a reflectivity of ~0.00003% was presented. This laser design ensures domination of dynamical population inversion grating over the FBGs in total distributed feedback, enabling laser stabilization and nonlinear filtering in the fiber cavity and the narrowing down of the laser's Lorentzian linewidth to ~290 Hz.

The implementation of a femtosecond (fs) laser refractive index modification technique for artificial disordered structures formed inside an optical fiber significantly simplifies the requirements for structure inscription accuracy compared with the periodic structures (FBG) inscription and allows one to reduce the length of the cavity when compared with the lasers based on natural Rayleigh backscattering [13]. Thus, in [14], a ring cavity laser with a random output reflector inscribed by a fs laser was performed. This RDFB structure was a set of planes with asymmetric cross-sectional index modulation and a thickness in the range of 6–10 $\mu$m, a depth of about 30–50 $\mu$m, and an arbitrary distance between neighboring planes in the interval of 10–30 $\mu$m. The sample had eight segments measuring 1 cm each, which, in turn, contained about 500 reflectors. Thus, the spectrum of a random reflector was complicated due to Fabry–Perot (as a result of core–core mode coupling) and Mach–Zehnder (as a result of core–cladding mode coupling) interferometer formation. Finally, multiple weakly reflective spectral filters ensured a single-frequency regime. The laser linewidth amounted to about 2.1 kHz at an output power of 2.9 mW.

In this paper, we report the realization of a narrow-linewidth $Er^{3+}$ fiber RDFB laser based on disordered structures produced using the femtosecond laser modification technique in a ring cavity configuration. A sample with R ~0.17% consisting of eight 12 cm long reflectors separated by ~40 cm sections of non-modified fiber was used in the experiment. A single-frequency generation regime was observed in the entire power range, for which the linewidth at a maximum output power of 7.8 mW did not exceed 0.75 kHz.

## 2. Experiment

To create a compact sample with an enhanced level of backscattering, the femtosecond laser modification technique was used [15]. Fs laser exposure leads to the formation of nanogratings inside a material with a typical crack thickness of ~10 nm and a period of ~100 nm [16] that allow for a significant enhancement of scattering. The effect of nanograting formation is explained by various mechanisms, ranging from the interference between the field of the fs laser pulse and the field of electron plasma formed during the pulse absorption [17] to the nanoplasmonic model [18].

For sample inscription, we used a Yb:KGW Light Conversion Pharos 6W (Vilnius, Litva) with the following parameters: a wavelength of 1026 nm, a pulse duration of 232 fs, a pulse energy of ~0.3 $\mu$J, and a repetition rate of 40 kHz. Femtosecond radiation was focused into the core region of a Fibercore SM1500SC(9/125)P (Southampton, UK) fiber using a Mitutoyo 50X Plan Apo NIR HR micro-objective (NA = 0.65) (Kanagawa, Japan). During

the process of inscription, the fiber was moved using Aerotech ABL1000 linear stage at a constant speed of 0.2 mm/s. To monitor the increase in the backscattering level resulting from refractive index modification and the formation of defects, we used a high-resolution LUNA OBR4600 backscatter reflectometer (Camarillo, CA, USA). The total length of the inscribed sample was ≈4 m. The sample consisted of eight reflectors with a length of 12 cm. The distance between neighboring reflectors varied from 40 to 50 cm, and the averaged induced backscattering level was +40–50 dB relative to the natural Rayleigh scattering level of the Fibercore SM1500SC(9/125)P (Southampton, UK) fiber depending on the concrete reflector (Figure 1a).

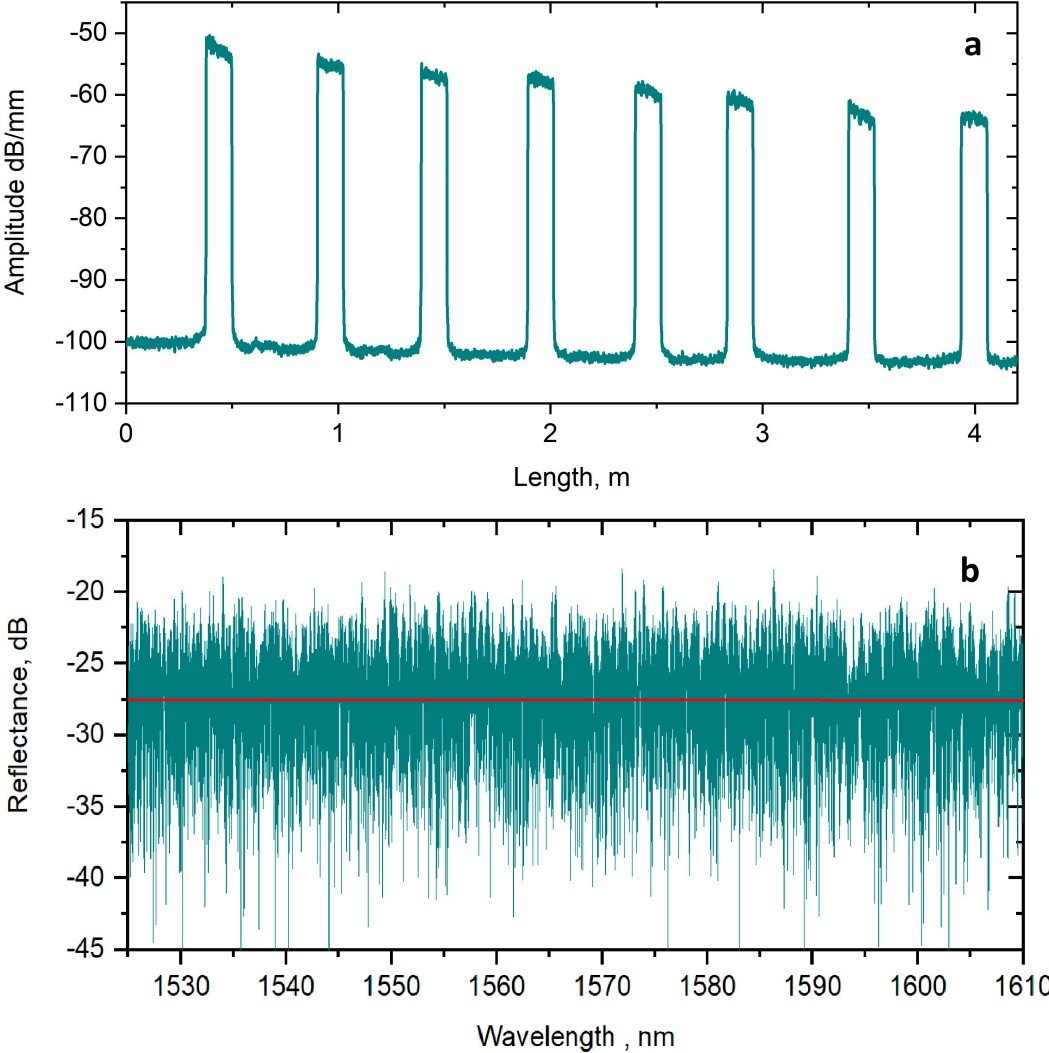

**Figure 1.** (**a**) Reflectogram and (**b**) reflection spectrum of RDFB structure.

The total reflectivity of the artificial reflector was 0.17% (Figure 1b), which is equal to the natural reflectivity of the Fibercore SM1500SC(9/125)P fiber with a length of ≈20 km. The level of the induced losses for each reflector did not exceed 15%. This was achieved by optimizing the inscription parameters, such as translation speed and pulse frequency and energy.

As an active medium, we used the fiber manufactured by the Fiber Optics Research Center (FORC RAS, Moscow, Russia). The fiber was made by sintering phosphate glass in a quartz glass tube and carrying out further drawing. The core and cladding diameters were about 4.5 and 125 μm, respectively. The manufacturing process is presented in more detail in [19]. For core manufacturing, glass with a composition defined in [20,21] was

used. Besides 65 mol. % of phosphorus oxide, this composition contained 7 mol. % of $Al_2O_3$, 12 mol. % of $B_2O_3$, 9 mol. % of $Li_2O$, and 7 mol. % of $Re_2O_3$ [22]. The erbium ion concentration was $1.6 \times 10^{20}$ cm$^{-3}$. This composition also contained gadolinium, and the total concentration of rare-earth ions was about 7 mol. %. It is important to note that the addition of aluminum oxide reduces the probability of clustering and leads to a continuous regime of laser operation even at high erbium ion concentrations. The signal absorption coefficients were 1.25 dB/cm @ 980 nm and 3.65 dB/cm @ 1535 nm. The loss level at the wavelength of 1300 nm was 3–5 dB/m, which was determined by the degree of impurity of the phosphate glass during processing. The gain coefficient for a weak signal was 3.1 dB/cm at 1535 nm.

The experimental scheme of the laser in a ring cavity configuration is presented in Figure 2. As a pump source, a 980 nm laser diode with a maximum output power of 500 mW was used. Pump radiation was supplied to the cavity through a WDM, and the length of the active fiber was 15 cm.

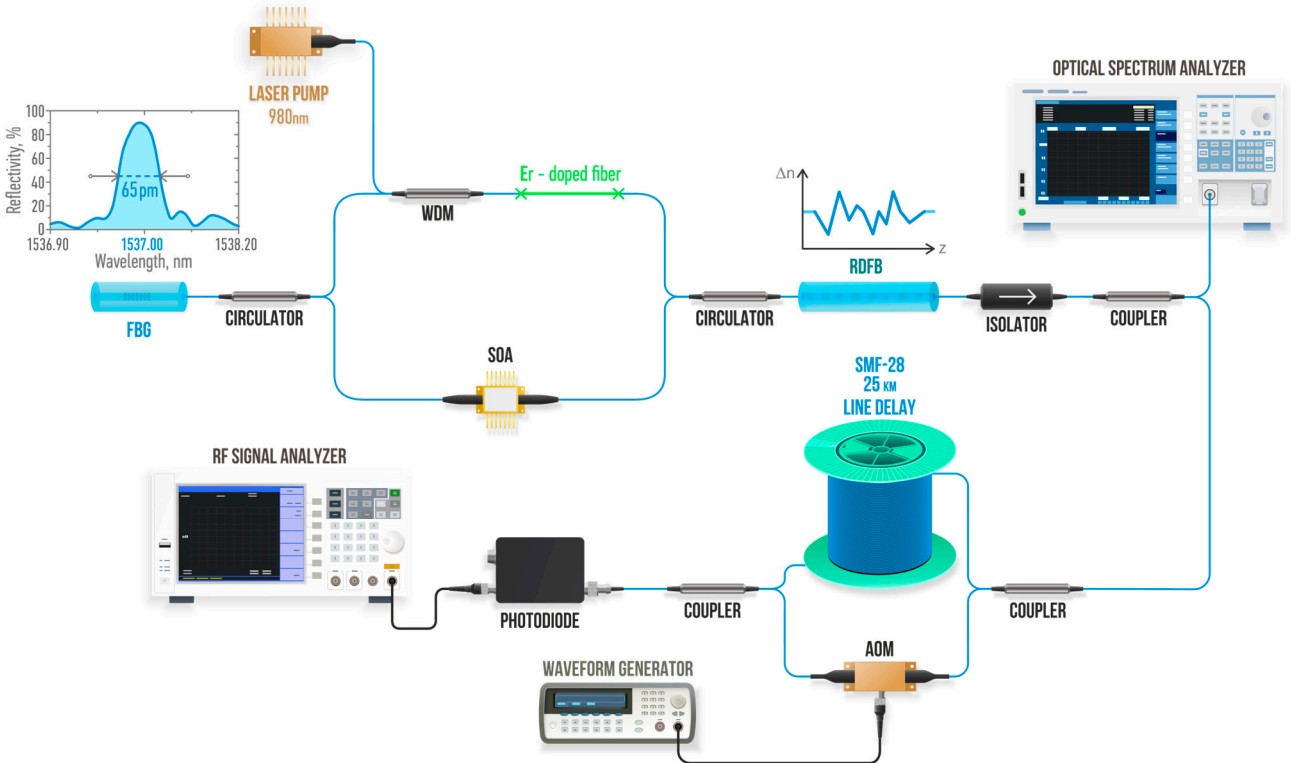

**Figure 2.** Experimental setup.

The artificial RDFB structure was incorporated into the ring cavity scheme via a circulator, as shown in Figure 2. In addition to a fiber amplifier, a semiconductor amplifier was used, providing broadband amplification. The signal reflected from the RDFB structure was amplified by a semiconductor amplifier with a coefficient of ≈16 dB at 1535 nm. After that, an FBG with a resonant wavelength of 1536.5 nm, a reflectivity of 95%, and a spectral width of 65 pm provided spectral filtering. As a result, narrow-linewidth generation arose from the combination of coherent Rayleigh backscattering in the RDFB structure, FBG filtering, and active fiber and semiconductor amplification.

To avoid back reflection, an optical isolator was placed at the output of the system. The spectral characteristics were measured together with output power using an optical spectrum analyzer (OSA), namely, a Yokogawa AQ6370 (Tokyo, Japan), with a resolution of 20 pm. Mode beating was measured with a 5 GHz Thorlabs DET08CFC (Newton, NJ, USA) photodiode and an Agilent N9010A radio frequency (RF) signal analyzer. For precise linewidth measurement, the self-heterodyne technique [23] was used. In this process,

one of the arms of the Mach–Zehnder interferometer contained a 25 km delay line, and another arm contained an acousto-optic modulator (AOM) powered by an Agilent 33250A waveform generator (Santa Clara, CA, USA).

## 3. Results

Figure 3a shows the dependence of the output power as a function of pump power. The lasing threshold was achieved at a pump power of 63 mW; at the maximum pump power of ≈500 mW, the output lasing power was 7.8 mW.

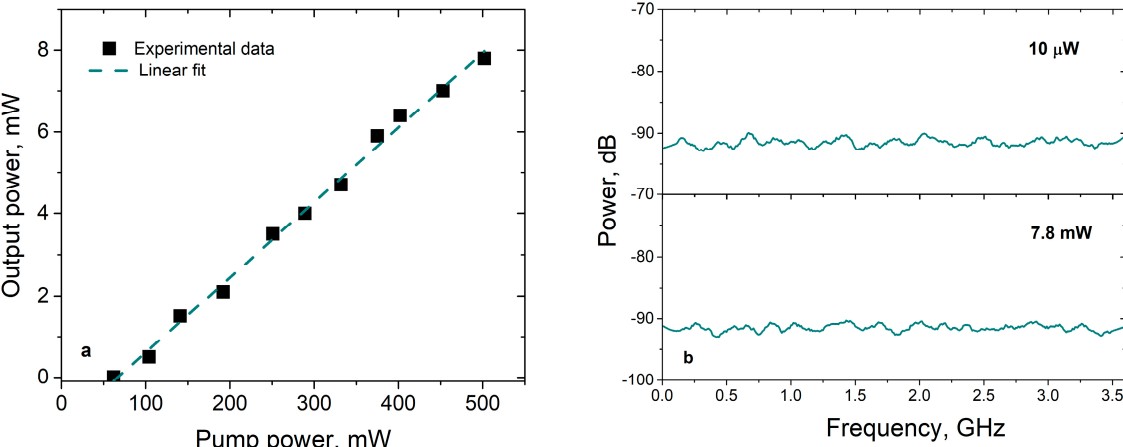

**Figure 3.** (**a**) The measured output power as a function of pump power; (**b**) RF spectrum at threshold and maximum output power.

Figure 3b shows the RF spectra of the signal near the threshold and at the maximum output power: there are no beating peaks of longitudinal modes in the entire power range, which corresponds to the single-frequency regime. Figure 4a shows the generation spectrum at the maximum output power: the central generation wavelength was about 1535.7 nm, the signal-to-noise ratio was 63 dB, and the full width at half maximum (FWHM linewidth) corresponded to the 20 pm resolution of the OSA.

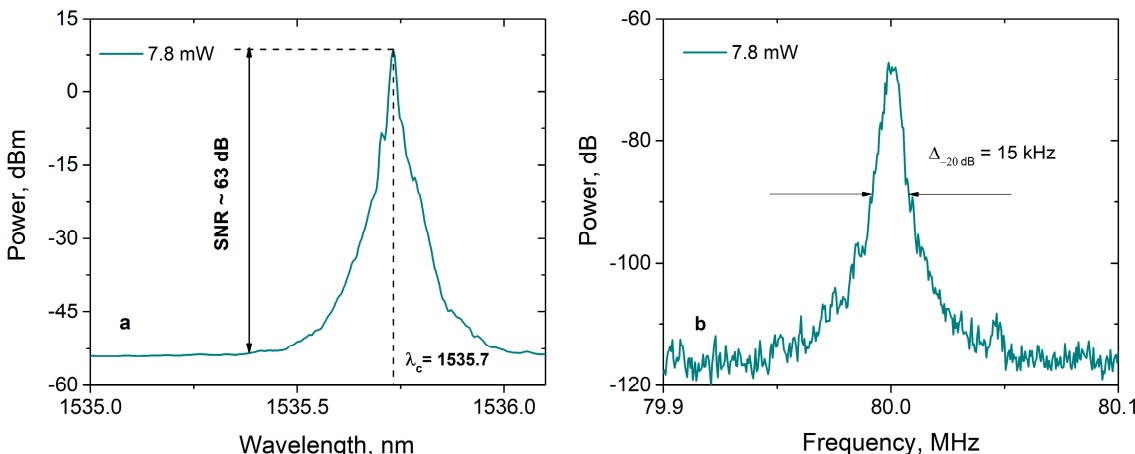

**Figure 4.** (**a**) Generation spectrum at maximum output power; (**b**) RF beating spectra obtained via heterodyne technique.

For the precise measurement of the linewidth, the self-heterodyne technique described in [23] was used. Figure 4b shows the beat spectrum at the maximum output power: the linewidth at the −20 dB level was 15 kHz, which corresponds to the laser linewidth at half maximum, ≈0.75 kHz [24].

The standard deviation of the output power for a time interval of ~10 min was about 2% (Figure 5a). The relative intensity noise at a frequency of 0.5 MHz corresponded to −105 dB/Hz (Figure 5b). Thus, the spectral characteristics of an RDFB laser in this configuration are not inferior to those of distributed feedback fiber lasers (DFB lasers) based on FBGs with a phase shift [25]. Moreover, in terms of output power, the presented laser outperforms erbium DFB lasers with a characteristic output power of ~100 µW by an order of magnitude.

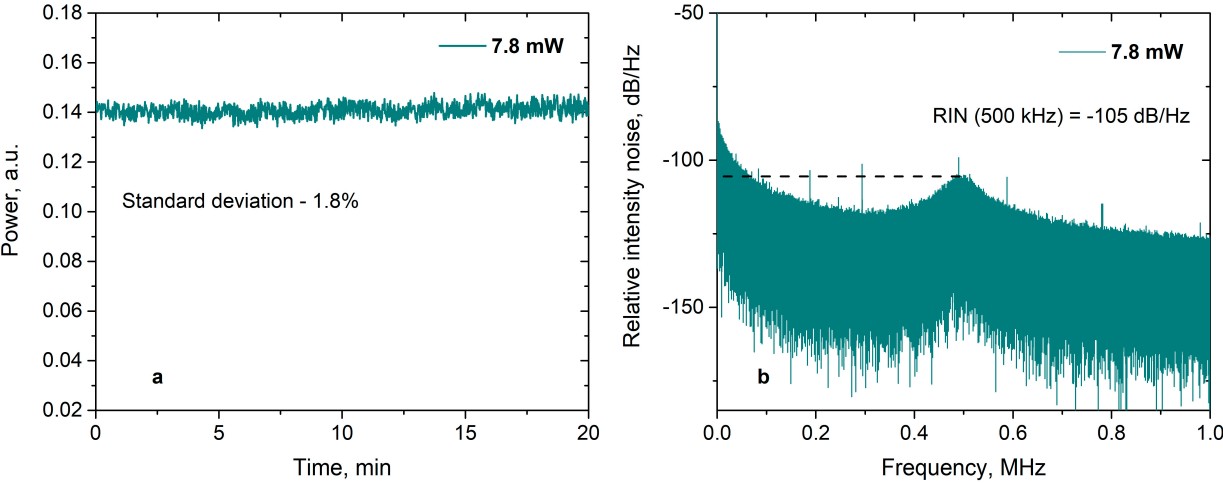

**Figure 5.** (**a**) Oscillogram of output radiation; (**b**) relative intensity noise measured at the maximum output power.

## 4. Discussion

Since RDFB provides elastic Rayleigh scattering feedback, the phase coherence is maintained between the incident waves and the waves reflected from the centers of the disordered medium. Scattered waves from the RDFB structure are amplified due to the fiber and semiconductor amplifiers in the resonator, and phase correlation leads to a complex interference pattern, which is confined by the FBG spectrum [7]:

$$E_{out} = \sum_{j=1}^{N} E_{ASE} \prod_{1}^{j} \exp\left(-\alpha L - i\frac{\pi n \nu L}{c}\right) \sum_{k=1}^{K} G_k R_k \exp\left(-2\alpha z_k - i\frac{4\pi n \nu z_k}{c}\right), \quad (1)$$

where $E_{ASE}$ is the amplified spontaneous emission, whose shape is determined by the FBG spectrum; $\alpha$ and $n$ are the loss coefficients and the refractive index of the optical fiber, respectively; $\nu$ and $c$ are the frequency of the electromagnetic wave and its speed of propagation in a vacuum, respectively; $L$ is the length of the ring resonator; $N$ is the total number of roundtrips of the composite resonator; $K$ is the total number of reflective RDFB centers; $z_k$ is the coordinate of the $k$-th center of the RDFB; and $R_k$ and $G_k$ are the coefficients of reflection and amplification of the electromagnetic wave for the $k$-th center of the RDFB structure, respectively. Figure 6 shows the simulated interference patterns depending on the number of roundtrips; these patterns were obtained using Equation (1): $I_{out} = E_{out}{}^* E_{out}$

Since the initial seed signal is an ASE from the fiber and semiconductor amplifiers with relatively low coherence determined by the spectral width of the FBG, only the input end of the RDFB structure provides coherent Rayleigh feedback. As the number of roundtrips of the resonator increases, the effective length of the RDFB reflector, capable of providing coherent reflection, increases due to the increase in the coherence length of photons as well as their power due to the use of fiber and semiconductor amplifiers. Since resonant peaks were observed due to the constructive interference of photons reflected from the RDFB structure, increasing the number of reflection centers leads to a higher probability of providing coherent feedback to amplify existing resonances.

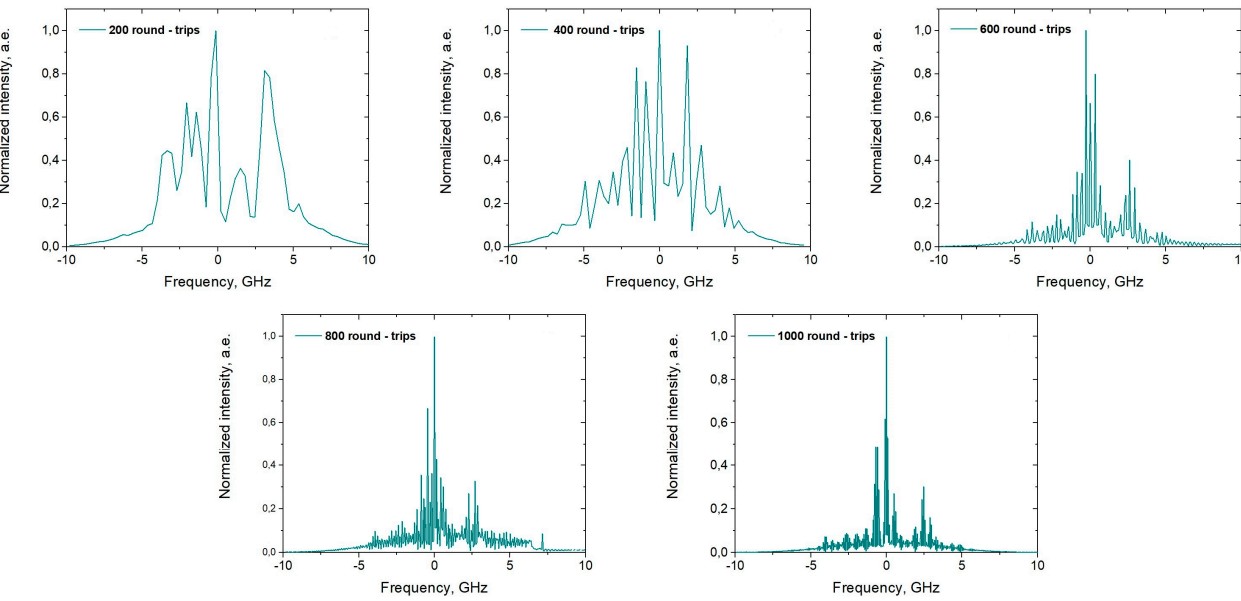

**Figure 6.** Simulated interference patterns of the proposed random fiber laser depending on the number of roundtrips.

Generation was observed at the frequencies for which the losses are compensated by the gain. The envelope of the gain spectrum is not flat-top, and it is determined by the FBG and narrows with an increase in the number of roundtrips; these factors finally lead to single-frequency generation (Figure 7).

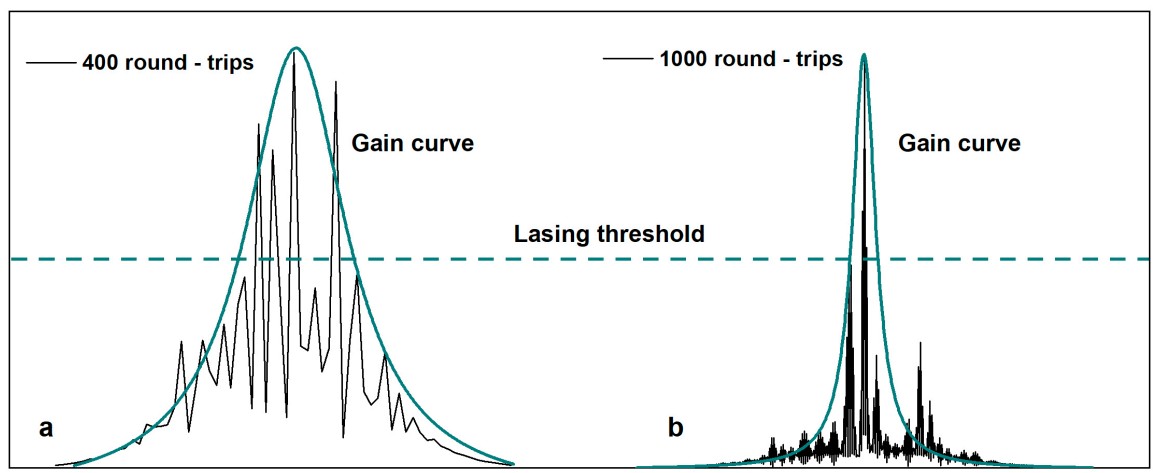

**Figure 7.** Schematic demonstration of longitudinal mode distribution in an RDFB laser cavity: (**a**) four modes and (**b**) only one mode exceeding the lasing threshold.

## 5. Conclusions

The implementation of the technique of refractive index modification via a fs laser makes it possible to form artificial disordered structures inside a fiber with a high level of induced backscattering. The requirements for the formation of such reflectors are noticeably simplified in comparison with the FBG inscription, and the length of the structure is several orders of magnitude less than the length of random reflectors based on natural Rayleigh scattering in optical fibers.

The use of this type of RDFB structure in an erbium fiber laser made it possible to obtain single-frequency lasing in a configuration with a ring cavity. The output power

was about 7.8 mW with a maximum pump power of $\approx$500 mW, and the signal-to-noise ratio was about 63 dB. The generation linewidth was 0.75 kHz, and the relative intensity noise at a frequency of 500 kHz was $-105$ dB/Hz. Thus, the spectral characteristics of the implemented RDFB laser are comparable to those of erbium DFB lasers [25], while the power level for the RDFB laser exceeds that of its counterpart by an order of magnitude, and the implementation of the experimental scheme is much simpler.

**6. Patents**

This study was supported by the state budget of IA&E (project FWNG-2024-0015).

**Author Contributions:** Conceptualization, S.A.B., S.L.S. and M.I.S.; data curation, S.A.B., M.I.S., V.S.T., A.A.W., Z.E.M. and K.V.P.; investigation, K.V.P., M.I.S., E.V.G., A.V.D. and V.S.T.; writing—review and editing, S.R.A., M.I.S., O.N.E., A.V.D., V.S.T. and S.A.B.; funding acquisition, S.A.B., S.L.S. and O.N.E.; formal analysis, E.V.G.; writing—original draft preparation, K.V.P.; project administration, S.A.B. and S.L.S.; supervision, O.N.E. All authors have read and agreed to the published version of the manuscript.

**Funding:** This study was supported by the state budget of IA&E (project FWNG-2024-0015).

**Institutional Review Board Statement:** Not applicable.

**Informed Consent Statement:** Not applicable.

**Data Availability Statement:** Data are contained within the article.

**Acknowledgments:** Experimental studies were carried out using the equipment of the Center for Collective Use "Spectroscopy and Optics" at the Institute of Automation and Electrometry, SB RAS.

**Conflicts of Interest:** The authors declare no conflicts of interest.

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
