# Peer review of "Single-Frequency Ring Fiber Laser with Random Distributed Feedback Provided by Artificial Rayleigh Scattering"

_photonics, doi:10.3390/photonics11020103_

Round 1
Reviewer 1 Report
Comments and Suggestions for Authors
The authors present a single-frequency ring fiber laser with random distributed feedback provided by fs laser induced random structures. The paper is technically sound and well written with overall good lasing performance in terms of output power, linewidth and stability. The paper can be accepted in Photonics after revisions.
(1) In introduction, some updated applications of random fiber laser in sensing and frequency conversion should be included, such as "Photonics Research 11 (5), 808-816", "Laser & Photonics Reviews, 2200797,2023, https://doi.org/10.1002/lpor.202200797"
(2) In the experimental setup, erbium-doped gain and SOA are used, so it is better to call their laser as hybrid gain fiber laser. What's the benefit to include the SOA gain, is it good for reducing the threshold or increasing the output power?
(3) In the linewidth measurement, they use 25 km delay fiber, which seems not long enough for measuring linewidth below 1kHz, why there are no interference patterns on the RF spectra?
(4)In Fig.5 (a), could the author provide the temporal intensity waveform in a short time scale, such as ms level?
Author Response
Dear reviewer,
thank you very much, your comments contributed to improving our work. Answers in the attached file

Reviewer 2 Report
Comments and Suggestions for Authors
The authors propose the use of fs laser to inscribe structures on optical fibers, then using the Rayleigh enhanced structures to implement a random distributed feedback (RDFB) fiber laser .
The implementation of the RDFB fiber laser using the Rayleigh enhanced optical fibers is novel. The authors provide a clear account of the method and experimental setting used to enhance the fiber and implement the RDFB fiber laser. The authors characterize the fiber laser operation fully.
Error on line 184.
My only suggestion is to enhance the quality of the figures. The images and text labels are pixelated.
Comments on the Quality of English Language
The manuscript is very well written.
Author Response
Dear reviewer,
thank you very much, your comments contributed to improving our work.
Reviewer 3 Report
Comments and Suggestions for Authors
The manuscript is well presented and the results are supported with a clear methodology. I'd like to arise few questions to the author to better improve the article.
1. I didn't get why do you need a Mach-Zender Interferometer (fig.2) to take out the signal since it is already acquired from an OSA.
2. what is it meant for "laser linewidth" which is mentioned in the article?
3. which is the novelty of this work compared with the state-of-art?
4. the section 4 seems to be slightly disjointed to the rest of the article. Which is the aim that leads the author to lead that analysis and what are the final goal achieved? please discuss better.
Author Response

(The authors gave the same response as above.)
